# Iterative Method of Discriminant Analysis to Classify Beech (*Fagus sylvatica* L.) Forest

**Alvaro Sánchez-Medina** [1,*], **Esperanza Ayuga-Téllez** [1], **Maria Angeles Grande-Ortiz** [2], **Concepción González-García** [2] and **Antonio García-Abril** [2]

1 Buildings, Infrastructures and Projects for Rural and Environmental Engineering (BIPREE), Universidad Politécnica de Madrid, Camino de las Moreras s/n, 28040 Madrid, Spain; esperanza.ayuga@upm.es
2 Methods & Technologies for Sustainable Management Research Group (MTSM), Technical School of Forest Engineering, Universidad Politécnica de Madrid, Camino de las Moreras s/n, 28040 Madrid, Spain; m.angeles.grande@upm.es (M.A.G.-O.); concepcion.gonzalez@upm.es (C.G.-G.); antonio.garcia.abril@upm.es (A.G.-A.)
* Correspondence: alvaro.sanchezdemedina@upm.es

**Abstract:** We present a new method for the classification of beech (*Fagus sylvatica* L.) forest plots based on discriminant and frequency analysis. This method can be used as a tool to allow experts to stratify beech forests in a simple and precise way. The method is based on discriminant analysis with cross-validation of 13 variables measured in 142 plots from the 2005 Second National Forest Inventory and 63 plots from an inventory installed in specific locations together with a frequency analysis of the qualifying variables. In the first stage, the method uses the results of a frequency analysis fitted with an iterative discriminant analysis that allows improving the subsequent classifications taking into account the results of the analysis and the correctly- and wrong-classified plots. This method is applied to beech forest in Burgos (Spain) where six structural groups were described. The discriminant functions show that forest structure depends basically on diameter distribution and almost 94% of the plots are correctly classified using this methodology. The high level of correctly assigned plots indicates an accurate classification of structure that can be used to stratify beech forests with only the diameter at breast height measurement.

**Keywords:** discriminant analysis; beech forest; diameter class; forest structure; forest management; multivariate statistics

## 1. Introduction

We introduce a new method that uses multivariate statistics to classify or stratify beech (*Fagus sylvatica* L.) forest plots into uniform structure types, thereby enabling the whole forest structure to be classified for the purpose of making decisions aimed at minimizing costs. This new method is based on discriminant analysis (DA) with cross-validation together with a frequency analysis (FA) of the qualifying variables that allows beech forest plots to be classified into structural types defined by dasometric variables.

Creating typologies of forest stands involves collecting together and synthesizing—using the same notation—stands that share certain characteristics considered decisive for the long-term objectives established and the silvicultural rules to be applied in the present. The typology of forest stands allows describing and identifying their structure; its usefulness is justified as the basis for a superficial diagnosis of the forest [1].

A number of studies—primarily in central Europe—have been carried out in order to define the structure and biological dynamics of beech stands, as a thorough knowledge of these forests is held to be essential for defining the appropriate forestry strategies as part of a close-to-nature forest management. The study of the structure affects a number of attributes of the forest stand. For example, regeneration in gaps of beech of different sizes imprint on the resulting canopy structure in different ways, enhancing spatial heterogeneity [2],

also variation in gap size is an important factor contributing to the composition of tree species composition of natural regeneration in beech-oak forests [3]. One of these attributes is the distribution of the *dbh* (diameter at breast high) of the trees, a technique frequently used by forest experts to describe particular forest types or silvicultural treatments [4] even. According to this distribution, forests can be classified into two major types: even- and uneven-aged. The first has an inverted U-shape diameter curve with a maximum at around the most abundant diameter range. Uneven-aged stands display a greater variability, with curves taking on a variety of shapes [5], including negative exponential shapes [6], inverted J-shaped curves [7], and diverse Weibull-type functions [8,9]. Alessandrini et al. [10] have made an in-depth study of this variability and found a trend towards a rotated sigmoid curve in virgin beech stands in the Italian Apennines. This same shape appears in other beech stands, such as in the Ukrainian forests of Uholka [7,11]. One of the points of attention is the size of the inventory plots. This is a relevant factor when extrapolating the data obtained from the analysis of structural typologies from plot to stand level. The plot size depends on the type of stand. It may be smaller in even-aged stands [12], or managed stands [13], than in old-growth or unmanaged beech forests.

The identification of structural typologies as a management tool cannot in some cases resolve certain problems arising from the heterogeneity found in this kind of beech forests, with their marked southern nature, although the heterogeneity appears not only in this latitude [14]. The plot sample size can only be reduced by locating it in representative zones in the stand; otherwise, a stand size similar to that determined by Alessandrini et al. [10] is recommended for uneven-aged stands. In studies on European virgin beech forests, sizes range from 0.1 ha [15], 0.25 ha [16], 0.5 ha [17] to 1.13 ha [18]. In any case, it is clear that multi-layered structures are extremely rare at the plot level, and become evident only on a wider scale [19].

The problem to conform structure typologies is basically to find silvicultural tools that allow minimizing management costs. Thus, the first structure typologies were purely descriptive, and gradually, multivariate statistics had been introduced with the aim to reduce field working and, on the other hand, to have a tool to measure the accuracy. A structural typology was developed earlier in the rest of Europe than in Spain. Herbet and Rebeirot [20] presented a descriptive classification for *Fagus sylvatica* L. and Abies alba Mill. forests in the Jura range based on data compiled from a hundred plots installed ad hoc. The work of Chollet and Kuss [21] on specific Pyrenean beech forests is also relevant, and already employs multivariate statistics, specifically a successive multiple correspondence analysis (MCA) to establish the different types. The identification of structural typologies has also taken place more recently in Spain than elsewhere. Of interest in Spain are the classifications made for some species: Roig et al. [22] used factorial analysis (FA) and cluster analysis (CA) to establish seven types of Juniperus thurifera L. forests and six types of stands of Quercus pyrenaica Willd; Aunós et al. [23] described nine groups for Abies alba Mill, using only FA, and Reque and Bravo [24] employed CA, FA, and DA to obtain nine groups for Quercus petraea (Matts.) Liebl. They obtained a discriminatory power from the DA of 87.5% (94.8% in the cross-validation). Specifically for beech, Gómez Manzanedo et al. [25] describe seven typologies for Cantabrian beech forests using FA, CA, and DA techniques. In this case, the discriminatory power was 80.6% (87.5% in the cross-validation). Except for Herbet and Rebeirot [20], all these works only use plots from the National Forest Inventory (NFI) and some multivariate analysis techniques [1], with a number of classificatory variables ranging between 5 and 20.

Other studies have also applied DA to study relationships between structure and tree competition [26] or how it affects silvicultural practices on the beech forest structure [27].

In this work we present a new method that has the next simultaneous improvements comparing with the methodologies related above: reducing the number of variables to measure and classify; high degree of accuracy (up to 90%); combining NFI plots with plots installed ad hoc in selected locations to compare results.

Through this study, we aimed to: (i) present the description of the method and its application to Burgos beech forests and (ii) analyze and discuss the outcomes of the DA in order to evaluate the possibilities of the proposed methodology.

## 2. Materials and Methods

We present a new method based on a DA that allows classifying beech plots in different structure typologies with a high degree of accuracy using as a variable the percentage of basal area of beech (% $BA_i$) distributed into four diameter classes ($DC_i$; $i$ = 1 to 4): diameter class number 1 ($DC_1$: 7.5 cm $\leq dbh \leq$ 12.4 cm); diameter class number 2 ($DC_2$: 12.5 cm $\leq dbh \leq$ 22.4 cm); diameter class number 3 ($DC_3$: 22.5 cm $\leq dbh \leq$ 42.4 cm); diameter class number 4 ($DC_4$: $dbh \geq$ 42.5 cm) and their relative distribution. The proposed method is based on an iterative process (Figure 1) in order to achieve an acceptable percentage of success in classified forest inventory plots according to their structure.

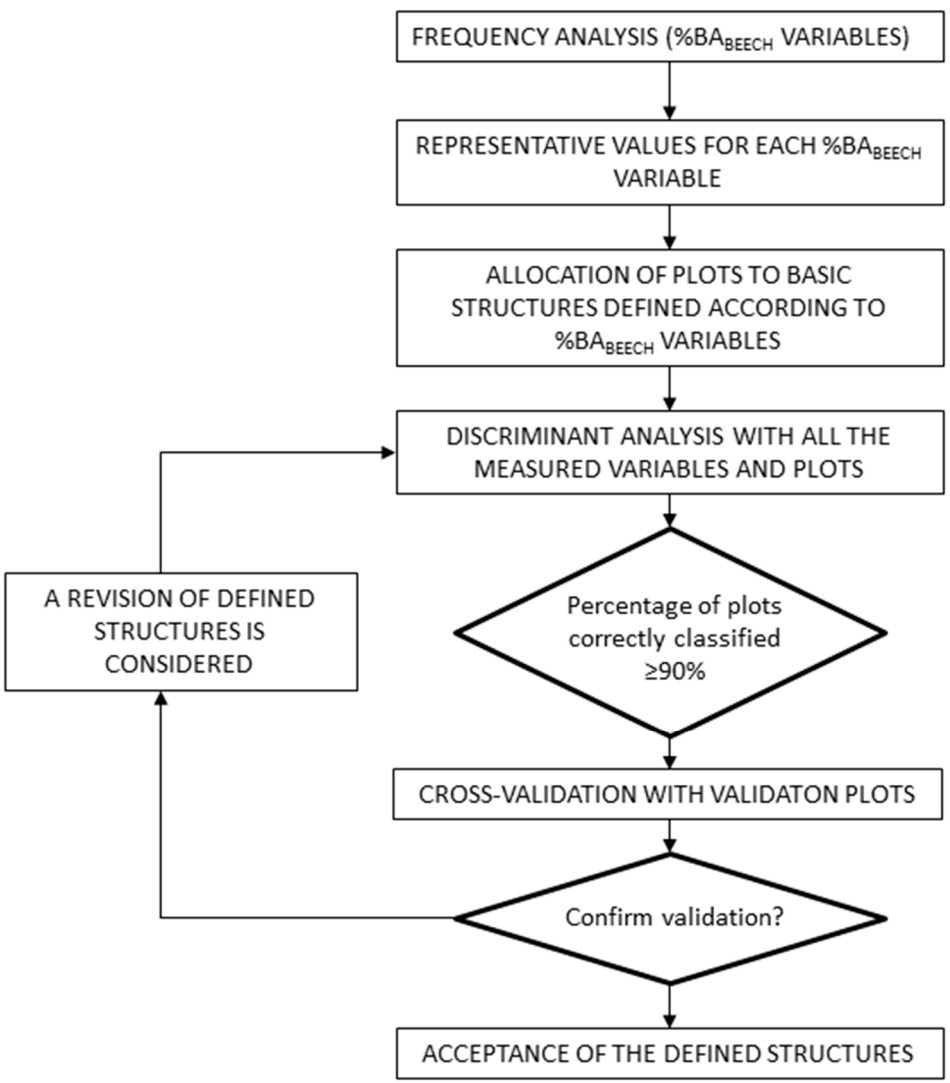

**Figure 1.** Flow process chart of the discriminant analysis method.

The method starts with a frequency analysis (FA) for each % $BA_i$ variable in order to determine the minimum values from which a diameter class can be considered as sufficiently represented (SR) related to its relative basal area, so the basic definition of structure typologies is based in that concept, also linked with crown surface and productivity [28]. Thus, it has been considered that a diameter class is sufficiently represented if it exceeds the value of the bottom quartile in each of the % $BA_i$ variables. The FA was performed

by taking all the plots with trees in all diameter classes and for each % $BA_i$ variable, the plots that did not have the maximum value in them. With the bottom quartile value, we can classify the plots considering the typologies detailed as follows. This categorization is based on an exhaustive exploration of the forests.

We consider three main categories: even-aged, uneven-aged, and stratified plots. Even-aged plots are those with a maximum of three continuous diameter classes sufficiently represented. Within even-aged ones, the Seedling Poles (*SP*) are those with the highest value in % $BA_1$ variable; Thin Poles (*TP*), those with the maximum value in % $BA_2$; Medium Poles (*MP*), those with the maximum value in % $BA_3$; and Thick Poles (*ThP*) those with the maximum value in % $BA_4$. In uneven-aged plots, all the diameter classes are sufficiently represented. Two subcategories could be differentiated: Multi-diameter (*MD*), those with not strictly uneven-aged values, and uneven-aged ss.ss. (*UA*) as those that have strictly uneven-aged values. The strictly uneven-aged values were obtained from data of four European uneven-aged beech stands: Uholka [7], Valle Cervara [10], Serrahn [16], and Kyjov [17], since there are no virgin forests in Spain. The ranges for each % $BA_i$ from the four diameter distributions were established with the average values $+/-$ the error (Table 1):

**Table 1.** Percentage ranges of basal area (% $BA_i$) for the strictly uneven-aged stand (average and error data from plots in Uholka, Valle Cervara, Serrahn and Kyjov).

| Variable | % $BA_1$ | % $BA_2$ | % $BA_3$ | % $BA_4$ |
|---|---|---|---|---|
| Average | 4.4 | 5.4 | 12.4 | 78.1 |
| Error | 2.8 | 3.0 | 3.7 | 2.7 |
| Range | 1.6–7.2 | 2.4–8.4 | 8.6–16.1 | 75.5–80.8 |

Finally, the Stratified plots (*S*) are defined as those with %$BA_2$ and/or %$BA_3$ are not sufficiently represented as is previously defined and one of the five following options occur (Table 2):

**Table 2.** Five cases for stratified plots (SR: Sufficiently represented; IR: Insufficiently represented).

| % $BA_1$ | % $BA_2$ | % $BA_3$ | % $BA_4$ |
|---|---|---|---|
| SR | IR | SR | SR |
| SR | IR | SR | IR |
| SR | IR | IR | SR |
| SR | SR | IR | SR |
| IR | SR | IR | SR |

In order to apply a DA, the Central Limit Theorem has been applied to assume the normality of the data. Once each plot has been classified, a DA with all the plots is made [29]. If the percentage of plots correctly assigned is below 90%, a revision of basic definitions is recommended, and so is a statistically acceptable confidence limit. Once this percentage is up to or equal to 90%, a cross-validation with some selected plots (validation plots or reserved sample) is made [30]. The significant variables were selected by the process of step-by-step inclusion, and the model was validated at each step by the *F*-statistic (*F* = 4 to entry, *F* = 4 to eliminate, tolerance = 0.001). The a priori probabilities were established by the percentage of cases in the sample. DA is made on the total data and the Fisher equations were calculated based on canonical correlation and the Wilks' Lambda distribution [31]. Once the percentage is up 90%, the definitions could be accepted if the percentage of success in the validation plots (reserved sample) is similar to the estimated sample.

In case to improve the structure definitions, two main results are obtained: the discriminant functions and the number of plots correctly- and wrong-classified. The j-discriminant function has the form:

$$Z_j = z_{j1}D_1 + z_{j2}D_2 + \cdots + z_{jp}D_p \tag{1}$$

where $D_p$ are the variables used to discriminate between different groups and $z_{jp}$, the weighting and standardized coefficients of each *p*-variable for each *j*-discriminant function.

These two results will be useful if a revision of the basic definition of typologies is considered. If this is so, the DA will be iterated with an improving definition. For this purpose, an exhaustive analysis of discriminant functions together with the wrong classified plots is recommended.

### 3. Application to Burgos Beech Forests

The method was applied to beech forests in Burgos (Spain). According to the Second National Forest Inventory (SNFI) [32], the Burgos beech forest occupies an extension of 170,662.46 ha, distributed mainly in two geographic zones: the northern area (101,648.97 ha; longitude: 4°20′ W–2°30′ W; latitude: 43°10′ N–42°30′ N) and the Demanda range (69,013.49 ha; longitude: 3°30′ W–2°45′ W; latitude: 42°20′ N–42°00′ N). These two areas have a continental climate with extremely cold winters and temperate summers with two months of drought. The average annual rainfall ranges from 500 to 800 mm, and the average temperature from 8 to 12 °C, with an average thermal oscillation of 28 °C. Beech forests grow above altitudes of 700 m in combination with other species, and up to 1500 m in pure or barely-mixed beech stands [33]. The structural variability of beech forests in this zone is widely diverse: even-aged and mono-specific stands at all development stages, multi-diameter stands mixed with pines and broad-leaved trees, pollarded trees with a dense regeneration stratum, etc.

In this region of Spain, there is roughly 14,000 ha of pure or mixed beech forest (considering the percentage of basal area for beech over 50%). This zone is distributed between the northern and eastern parts of the region, in stands with areas of between 30 and 250 ha. Beech forests occupy almost 12% of the total surface area of the Burgos region, most of which is destined for public utility. According to the Second National Forest Inventory (SNFI) [32], beech forests have increased in area by nearly 60% since the First National Forest Inventory (FNFI) [34], particularly trees with a *dbh* $\leq$ 12.4 cm, with almost three times the stock. This increase is evidence of the local significance of beech as a species in forest management, where a prior diagnosis of this wooded formation is a key to an understanding of the increasing importance of this species in recent decades [33]. In spite of its productive significance, beech in this region has been considered a species with a secondary role and has tended to be used only as non-commercial wood. It is thus essential to apply specific management practices for beech in order to protect and preserve forests that require special attention due to their environmental value, and—in some locations where current growth is higher than the average—to their productive potential. This type of management requires a specific tool to define the kind of beech structure that is typical of this area.

Regarding forest structure, Burgos beech forests can be divided into two large groups: on the one hand, even-aged stands with different degrees of evolution; and on the other, stands with a wide range of diameter classes that could be termed uneven-aged. According to the 2005 SNFI, they are all barely productive (average basal area (*BA*): 14.92 m$^2$; current growth: 1.98 m$^3$/ha). There are no stands with the specific characteristics of a virgin forest, as can be found in central Europe. This is seen by comparing the data from inventory plots with average data from four selected central European unmanaged forests: Uholka [7], Valle Cervara [10], Serrahn [16], and Kyjov [17,35]. All of the uneven-aged stands studied in central European virgin beech forests have similar characteristics in relation to the distribution of the *dbh* and the BA. The degree of mixture with other species is fairly limited (in general, these are stands where beech occupies more than 90% of the cover). All diameter classes are represented. The BA is high, with averages of over 35 m$^2$/ha, and the greatest percentages in the higher diameter classes, in trees with a *dbh* of over 40 cm. There is a high percentage of deadwood, in some cases up to 10% of the total trees in the stand. There are no stands in the study area with the same specific characteristics such as a central European unmanaged beech forest.

This work was compiled using two data sources: selected plots from the SNFI and a sampling of plots of different structures in representative locations that characterize the variability of forest structures.

The SNFI is a stratified sampling with strata formed by groupings of forestry areas with similar features. A sample distribution of these strata was made, with a proportional allocation to their area, obtaining a sample intensity of one plot every 100 ha. The plots have variable radii with 5, 10, 15, and 25 m, and trees with a minimum diameter of 7.5, 12.5, 22.5, and 42.5 cm respectively were measured in these areas. We selected plots with beech as the main species, and inside this selection, the plots in which the BA of beech is higher than 50% of the total BA of the plot. In this way, 142 plots were obtained.

On the other hand, the field sampling made in 2005 was conducted with a twofold goal: firstly, to increase the sample size, and secondly to use it as a reference and verification for the results obtained on 2005 SNFI plots. This sampling was made in nearby representative areas of the forest stratum established in the 2005 SNFI, at a distance of less than 100 m from the SFNI plots, and increased the sample size by 44.3%. The plots were selected in such a way that they shared the diametric distribution characteristics of the SFNI plots. In total, we sampled additionally 63 square plots of 400 m², selected based on two criteria: one, the plots had to be located near plots in the 2005 SNFI; and second, these plots had to be representative of the strata defined. The total number of plots selected from both inventories was 205. This represents an inventory intensity of 0.2% in relation to the total area of beech forest, a higher order of magnitude than in other similar works [36–41].

Plot size can thus be substantially reduced, as the plots are identified in such a way as to be the most representative of the stands to which they belong. As an example, Chollet and Kuus [21] work with plots representing the structural variability of beech stands with an area of about 0.07 ha. This work included exploratory visits to the zone accompanied by forestry technicians and experts.

The trees were divided into the four diameter classes as were commented above. These levels or diameter classes are the same as used in the adapted radius plots in the 2005 SNFI. The variables used to identify the structures were: the number of stems per hectare (*N*), Assman dominant height (*HDOM*) defined as the mean height of the hundred largest trees per hectare, percentage of BA per diameter class (% $BA_i$: *i* = 1 to 4), percentage of BA of the rest of species per diameter class, (% $BAREST_i$: *i* = 1 to 4), total *BA* (*BATOT*), beech *BA* (*BA*), percentage of *BA* for beech (% *BA*), beech regeneration (*REGE*), and regeneration for the rest of the species (*REGEREST*). The last two variables are estimates of the number of seedlings. *BA* was used instead of *dbh*, because *BA* better-emphasized diameter classes that predominate in each structure, both visually and in timber volume.

For sample sizes over 120 plots [42], cross-validation with division into only two groups is similar to validation obtained with other methods. Thus with a sample of 205 plots, the first step is to divide the sample into two groups: one part is used for the analysis and the other part is withheld to validate the results. The minimum size of the sample for validation is determined depending on the predictor variables used, the number of groups, and the canonical correlations of the model [43]. The results for the groups, variables, and correlations [44] in the first DA indicated that a minimum of 60 plots was necessary.

Although numerous works on agroforestry apply a 50% division of the sample for validation [31,45], the recent work of Ramayah et al. [46] recommends withholding 35% of the sample. Conversely, according to Neville et al. [47], a sample withholding of over 30% decreases the goodness of classification. Some studies use 25% of the withheld sample for validation [48,49]. In the work of Reque and Bravo [24], only 10% of plots were used for validation (fewer than 30 plots).

For the present work, 205 plots were used to obtain the discriminant functions, from which 60 plots were withheld for cross-validation. This is the minimum value for validation and is less than 30% of the plots (60 plots selected by means of stratified sampling) where strata are the typologies, and within each stratum, systematic sampling is made with

proportional allocation [50]. This size of the withheld sample fulfills the conditions required in previous studies.

## 4. Results

For each variable, a box plot has been obtained to show its bottom quartile (Figure 2).

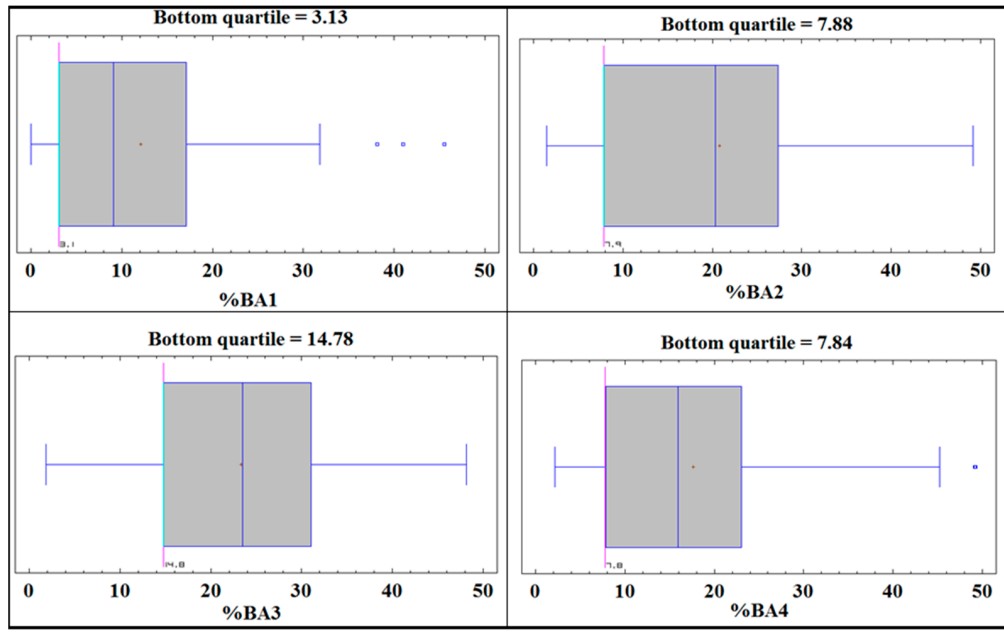

**Figure 2.** Box-and-whisker plot for the frequency analysis for each % $BA_i$ variable.

With these values, once the plots were classified in their respective groups, a first discriminant analysis was made with all the variables measured. The two first discriminant functions ($Z_1$, $Z_2$) had the following expression:

$$Z_1 = 0.93 \times (\%BA_1) + 0.39 \times (\%BA_2) + 0.22 \times (\%BA_4) \tag{2}$$

$$Z_2 = 0.53 \times (\%BA_1) - 0.14 \times (\%BA_2) + 0.86 \times (\%BA_4) \tag{3}$$

The variables included in the discriminant functions for the classification of six beech typologies were: % $BA_1$, % $BA_2$ and % $BA_4$. These variables produced two functions with canonical correlations over 0.7 and significant and low values of Wilks' Lambda (*p*-value = 0) in all cases. The Fisher classification functions for this first DA are shown in Table 3.

**Table 3.** Fisher classification functions for first DA (*SP*: Seedling Poles; *TP*: Thin Poles; *MP*: Medium Poles; *Th*: Thick Poles; *S*: Stratified; *MD*: Multi-diameter).

| Variable | Group | | | | | |
|---|---|---|---|---|---|---|
| | *SP* | *TP* | *MP* | *ThP* | *S* | *MD* |
| % $BA_1$ | 0.569 | 0.267 | 0.092 | 0.130 | 0.217 | 0.197 |
| % $BA_2$ | 0.302 | 0.462 | 0.160 | 0.186 | 0.168 | 0.270 |
| % $BA_4$ | 0.155 | 0.154 | 0.087 | 0.366 | 0.256 | 0.180 |
| Constant | −25.795 | −17.915 | −3.097 | −16.577 | −10.673 | −9.059 |

The percentage of plots correctly assigned with this first classification was 79.6%, with the worst results corresponding to the plots with an uneven-aged structure, in which no more than 50% were correctly assigned. No plot was classified as uneven-aged (*UA*).

Of the 41 plots wrongly classified with the first discriminant function, 61% (25 plots) were correctly assigned with the second discriminant function ($Z_2$), most of them corresponding to multi-diameter (*MD*) plots.

Related to even-aged ones, all the plots correctly classified were those that had the percentage of basal area that defines each typology over 50% (% $BA_1$ for *SP*, % $BA_2$ for *TP*, % $BA_3$ for *MP*, and % $BA_4$ for *ThP*). This assumes an average of 89.86% of correctly classified cases in even-aged plots (86.21% for *SP*, 88.00% for *TP*, 97.26% for *MP*, and 80.00% for *ThP*).

For stratified (*S*) and multi-diameter (*MD*) plots, we obtained 40.91% and 48.15% of correctly classified plots, respectively. The comparison between percentages of correctly- and wrong-classified plots with this first DA is shown in Figure 3.

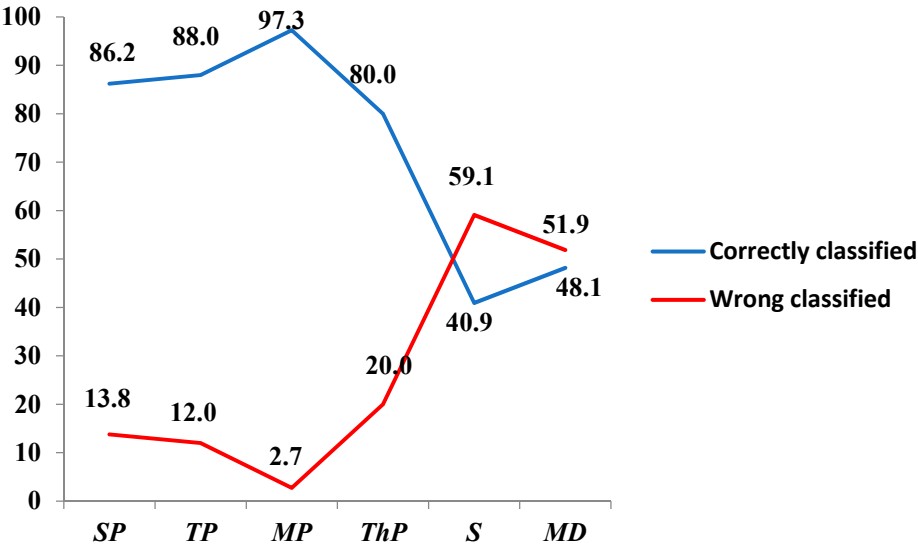

**Figure 3.** Percentage of correctly- and wrong-classified plots with the first DA. (Legend: *SP*: seedling poles; *TP*: thin poles; *MP*: medium poles; *ThP*: thick poles; *S*: stratified; *MD*: multi-diameter).

Taking these results as the basis for improving the classification, new assignation guidelines were detected. A second classification was made with all these assumptions as follows: we considered a new categorization of structures:

The Seedling Poles (*SP*) is the one with % $BA_1 \geq 50$ and does not have the condition of stratified.

Thin Poles (*TP*), is the one with % $BA_2 \geq 50$.

Medium Poles (*MP*), those with % $BA_3 \geq 50$ and does not have the condition of stratified.

Thick Poles (*ThP*) is the one with % $BA_4 \geq 50$ and does not have the condition of stratified.

Stratified (*S*) is the one with % $BA_2 = 0$ and simultaneously, the difference between (% $BA_3$ + % $BA_4$) and % $BA_1$ in absolute value is less to 20%.

Multi-diameter (*MD*) was defined by exclusion, as the one without any of the previous conditions.

The plots were reclassified and a new DA was made. The statistical program gave the following expression for the two first discriminant functions $Z_1$ and $Z_2$:

$$Z_1 = 1.14 \times (\%BA_1) + 0.55 \times (\%BA_2) + 0.42 \times (\%BA_4) \tag{4}$$

$$Z_2 = 0.17 \times (\%BA_1) - 0.16 \times (\%BA_2) + 1.10 \times (\%BA_4) \tag{5}$$

In this case, the variables included in the discriminant functions for the classification of six beech typologies were the same as before: % $BA_1$, % $BA_2$, and % $BA_4$. These

variables produced three functions with canonical correlations over 0.7 and significant and low values of Wilks' Lambda (*p*-value equal to 0) in all cases. The coefficients of Fisher classification functions for the second DA are shown in Table 4.

**Table 4.** Coefficients of the Fisher classification functions for second DA (*SP*: Seedling Poles; *TP*: Thin Poles; *MP*: Medium Poles; *Th*: Thick Poles; *S*: Stratified; *MD*: Multi-diameter).

| Variable | Group | | | | | |
|---|---|---|---|---|---|---|
| | *SP* | *TP* | *MP* | *ThP* | *S* | *MD* |
| % $BA_1$ | 0.857 | 0.450 | 0.146 | 0.369 | 0.647 | 0.373 |
| % $BA_2$ | 0.478 | 0.705 | 0.206 | 0.389 | 0.361 | 0.448 |
| % $BA_4$ | 0.386 | 0.353 | 0.157 | 0.604 | 0.460 | 0.337 |
| Constant | −39.684 | −29.540 | −3.658 | −26.587 | −29.140 | −14.640 |

The successful classification of the plots selected with the preliminary DA (145 plots) was 91.5%, and 95.0% for the non-selected plots used for the validation (60 plots). The model's level of success was very high, as in both cases the validation correctly classified over 90%. The percentage of correctly classified plots with this second DA with the cross-validation is shown in Figure 4.

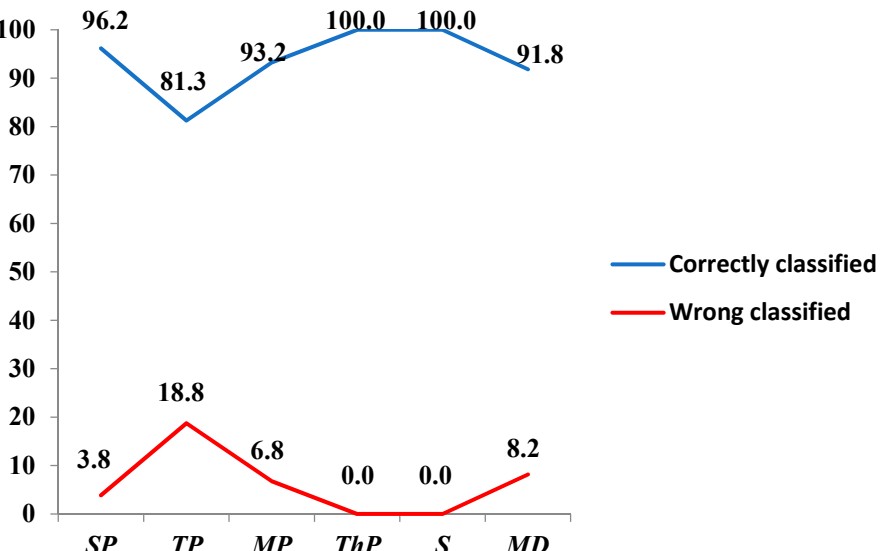

**Figure 4.** Percentage of correctly- and wrong-classified plots with the second DA with cross-validation. (Legend: *SP*: seedling poles; *TP*: thin poles; *MP*: medium poles; *ThP*: thick poles; *S*: stratified; *MD*: multi-diameter).

There are two classes with 100% correctly classified cases: *SP* and *ThP*, while *TP* is the group with the least satisfactory result (81.25% of plots correctly classified). Considering all typologies, the total percentage of correctly classified plots is 93.75%. This result is similar to that obtained for the Cantabrian range for sessile oak [24], with a discriminatory power of 87.5% accuracy and 94.5% in the cross-validation.

Of the total plots (205), 26.3% are reclassified in other groups in the Second DA (Table 5). It is observed that the highest percentage of reclassified plots is the MD group with 10%, followed by the *S* group with 5%. *TP* and *ThP* present a similar percentage, 4.4%, and the groups in which less reclassification occurs are in *SP* with 1.5% and in *MP*, which reclassifies the same in almost 100% of cases (only 0.5% of cases are reclassified).

**Table 5.** The number of plots for each discriminant analysis for each group (*SP*: Seedling Poles; *TP*: Thin Poles; *MP*: Medium Poles; *Th*: Thick Poles; *S*: Stratified; *MD*: Multi-diameter).

| DA | Group | | | | | |
|---|---|---|---|---|---|---|
| | *SP* | *TP* | *MP* | *ThP* | *S* | *MD* |
| First DA | 26 | 16 | 74 | 34 | 2 | 49 |
| Second DA | 29 | 25 | 73 | 25 | 22 | 27 |

By groups, the reclassification is distributed as follows:

- *SP*: 1 plot reclassified as *S*
- *TP*: 1 plot reclassified as *MD*
- *MP*: 5 plots reclassified as S, 5 plots reclassified as *MD*.
- *ThP*: 13 plots reclassified as *S*
- *S*: stay the same
- *MD*: 4 plots reclassified as *SP*, 10 plots reclassified as *TP*, 9 plots reclassified as *MP*, 4 plots reclassified as *ThP* and 1 plot reclassified as *S.*

To have a description of each typology attending to the discriminant variables, Table 6 shows the mean value the standard deviation for the discriminant variables for each group.

**Table 6.** Results of the discriminant variables for each typology: the mean and standard variation by group value (*SP*: Seedling Poles; *TP*: Thin Poles; *MP*: Medium Poles; *Th*: Thick Poles; *S*: Stratified; *MD*: Multi-diameter).

| Variable | Mean | | | | | |
|---|---|---|---|---|---|---|
| | *SP* | *TP* | *MP* | *ThP* | *S* | *MD* |
| % $BA_1$ | 0.569 | 0.267 | 0.092 | 0.130 | 0.217 | 0.197 |
| % $BA_2$ | 0.302 | 0.462 | 0.160 | 0.186 | 0.168 | 0.270 |
| % $BA_4$ | 0.155 | 0.154 | 0.087 | 0.366 | 0.256 | 0.180 |
| Constant | −25.795 | −17.915 | −3.097 | −16.577 | −10.673 | −9.059 |

| Variable | Standard Deviation | | | | | |
|---|---|---|---|---|---|---|
| | *SP* | *TP* | *MP* | *ThP* | *S* | *MD* |
| % $BA_1$ | 0.569 | 0.267 | 0.092 | 0.130 | 0.217 | 0.197 |
| % $BA_2$ | 0.302 | 0.462 | 0.160 | 0.186 | 0.168 | 0.270 |
| % $BA_4$ | 0.155 | 0.154 | 0.087 | 0.366 | 0.256 | 0.180 |
| Constant | −25.795 | −17.915 | −3.097 | −16.577 | −10.673 | −9.059 |

## 5. Discussion

The results of the first DA determined the improvement of the classification. In the case of stratified ones, most are classified in *MP* or *ThP* typologies so they have an outstanding value over 70% in % $BA_3$ or in % $BA_4$, respectively. In all these cases an unbalance between the two strata is plain. Stratification—or two-layered stands—implies the lack of an intermediate stratum that rises high enough for the dominant and co-dominant stratum. This niche is occupied by the lower stratum, which develops under the protection and coverage of mature trees. Most of the stratified plots have no trees with a thin pole size, and the arithmetic difference between the percentages of medium and thick poles and seedling poles is always lower than 20%; this means that there must be a balance between the dominant stratum and the stratum developing under its cover. Looking into the correctly assigned ones, the pattern seems to be the balance between dominant and dominated strata. Regarding *MD* plots, almost 85% of the wrong classified plots were correctly classified with the $Z_2$ function, so this typology guideline seems not to be misleading. The main unifying feature of multi-diameter stands is that they present continuity in all developmental ages or diameter classes. There are cases with an absence of some of the extreme types of wood:

for example, there could be a lack of seedling poles (mature and truncated multi-diameter stand) or thick poles (young and truncated multi-diameter stand).

The discriminant functions for both DA, take only % $BA_1$, % $BA_2$, and % $BA_4$ as discriminant variables. That shows that the stand structure basically depends on its diameter distribution. This result is consistent with those obtained by Bannister and Donoso [51] where it is suggested that the basal area of forest can be a relevant variable for forest typification. This is also important to define sustainable silvicultural alternatives. The DA highlights the greater importance of % $BA_1$ and % $BA_2$ variables in the classification, that is, the presence or absence of young trees. % $BA_i$ is an adequate variable to describe forest structure not only because it appears as a discriminant variable but also because it represents in a good way the amount of wood within a diameter class.

We obtained an increase of 13.6% in the average percentage of the correctly classified plots. In spite of the percentage obtained with the first DA, this could be considered acceptable since the low value for *S* and *MD* typologies does not show this satisfactory. Thus, the second definition of typologies is more restrictive for stratified plots and this implies an increase in the percentage of correctly classified plots of this type as well as a reduction of the total number of them.

The *TP* typology has the lowest mean value in its diameter class compared to the rest of the even-aged stands. This could be an important factor in explaining the low relative percentage of correctly assigned cases for the first DA. These stands represent a development stage in the transition from *SP*, or towards MD. They are not yet sufficiently mature, and this is the stage of the highest competition. In the other three types of even-aged stands, the percentage of the BA of the dominant diameter class is always over 70%, with percentages of correctly assigned plots of between 95% and 100%.

Stratified stands are very scarcely represented. It is unusual to find a well-formed stratification that presents a clear diameter discontinuity. In these stands, there is a considerable amount of bare ground and no species other than beech in the regeneration stage. Beech grows much better than other species under these conditions of isolation. This occurs with beech in its southernmost distribution, where the periodical small-gap creation implies an advantage in the regeneration of beech compared to other species [52]. This is also important to improve silvicultural treatments, as can be reviewed in [53] and [26] where BA is considered one of the most sensitive variables to those treatments.

## 6. Conclusions

The new methodology proposed in this work gets statistically satisfactory results (≥90%) to classify beech forest plots defined through the percentage of BA. The field sample provides a specific structural stratification based on the stratum in the NFI. This allows structures to be identified in the field and thus helps to produce a classification that is more closely aligned with reality. It is therefore recommended to use the BA as a key variable in the definition of beech forest structures. The proposed method provides a tool to estimate the percentage of each typology in order to improve logistical and economic planning. Also, it is useful to determine the silvicultural treatments to be applied.

**Author Contributions:** Conceptualization, A.S.-M., A.G.-A. and E.A.-T.; methodology, A.S.-M. and E.A.-T.; software, A.S.-M.; validation, C.G.-G. and M.A.G.-O.; formal analysis, A.S.-M. and E.A.-T.; investigation, A.S.-M., A.G.-A., and E.A.-T.; resources, C.G.-G.; data curation, A.S.-M.; writing—original draft preparation, A.S.-M.; writing—review and editing, A.S.-M., M.A.G.-O., E.A.-T., C.G.-G. and A.G.-A.; visualization, M.A.G.-O.; supervision, C.G.-G.; project administration, C.G.-G. All authors have read and agreed to the published version of the manuscript.

**Funding:** This research received no external funding.

**Institutional Review Board Statement:** Not applicable.

**Informed Consent Statement:** Not applicable.

**Data Availability Statement:** The data presented in this study are available on request from the corresponding author.

**Conflicts of Interest:** The authors declare no conflict of interest.

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
