# Peer review of "Iterative Method of Discriminant Analysis to Classify Beech (Fagus sylvatica L.) Forest"

_forests, doi:10.3390/f12081128_

Round 1
Reviewer 1 Report
Title: Iterative method of discriminant analysis to classify beech (Fagus sylvatica l.) forest
Abstract: This section is well written and captures the entire study.
Introduction:
I suggest deleting line 105 to 111 and instead give a brief introduction for the various chapters
Materials and Methods:
I find this section well-structured and scientifically valid
The discussion section mainly focusses on the present study. I will suggest to compare the study’s outcome with similar studies. You may as well merge the results section to the discussion section to enhance the flow.
The conclusion section needs to be more concise and widened and some recommendations provided.
In general, I find the study very important as it bring to the fore a new classification method for forest tree species to aid in cost cutting. The study was also well structured and will appeal to readers. I will only suggest concerns raised are addressed and the entire study check for any grammar error.
Author Response
- We have compared the study's outcomes with other studies.
- In relation to joining the discussion with the results, we have not considered it appropriate, since the results are quite extensive and could confuse the reader.
- We have rewritten the conclusion section more concise and introduciong some recommendations to use the proposed methodology.
- We have checked the format and grammar again.
Reviewer 2 Report
Dear Authors,
I think the paper is all right and could be published.
Regards
Author Response
OK.
This manuscript is a resubmission of an earlier submission. The following is a list of the peer review reports and author responses from that submission.
Round 1
Reviewer 1 Report
The aim of the manuscript »Iterative method of discriminant analysis to classify beech (Fagus sylvatica l.) forest« is to present a new method for classification of beech forest plots based on discriminant and frequency analysis. The method is to be used as a tool to allow experts to stratify beech forests in a simple and precise way.
The data and sample design are derived from the Spanish national forest inventory (SNFI), which could be a good starting point for assessing and possibly applying the proposed method in other countries for which national inventories have already been harmonized. Unfortunately, the authors did not describe the key features of SNFI to make the working methods understandable even to those researchers who are not involved in the work in the field of national forest inventories. Thus, the key question of why the presented diameter (dbh) classes were selected and what the dbh classes of 7.5–12.4 cm, 12.5–22.4 cm, 22.5–42.4 cm, and above 42.5 cm represent, remains unexplained. The same is true for the description of probably concentric circular plots in SNFI, with radii of 5, 10, 15, and 25 m. It can be concluded that the sample plots ranged from 78 m2 to 0.19 ha, probably with very different plot sizes, depending on the dbh structure of the trees on the individual sample plots. The design of sampling is therefore difficult to compare with other studies in the introductory chapter where fixed sample plot sizes were used. It is written in the chapter on conclusions that the stand structure basically depends on its diameter distribution. This was not analyzed at all in the manuscript. Only the percentages of basal area (BA) by dbh classes were used in the analysis. It can be concluded that the classification was adapted to the data and was not based on any substantive reasons on the basis of which the classes of BA on the sample plots would be determined. The results are thus limited to the sample plots in Burgos (Spain) and cannot be estimated, assumed and compared in similar surveys of other beech forests. The discussion chapter was not presented by the authors at all.
The manuscript is written inaccurately in describing variables and carelessly in stating units of measurement.
More specific comments are described below.
Lines 50-54. Awkwardly written statement in which the horizontal distribution of the trees and dbh distibution are mixed. … the interpretation of the horizontal distribution of the trees, a technique frequently used by forest experts to describe particular forest types or silvicultural treatments [4] even. According to this distribution, forests can be classified into two major types: even- and uneven-aged. The first have an inverted U-shape diameter curve with a maximum in around the most abundant diameter range.
L 133. Even-aged plots are (?) those with …
L 144, 151, 290 (Figure 2), 296, 297, 350, 357, 358. Unify the labels of the variables. Write the variable label the same everywhere, with a space (% Bai).
L 187-188. Degrees of latitude and longitude.
L 191-192. 0C
L 202. Describe tree sizes.… beech forest has increased its area by nearly 60% in all tree sizes.
L 218, 225, 245. m2, m3/ha.
L 220-228. The virgin forests cannot be inferred solely from the amount of BA, diameter classes and amount of dead wood.
L 237. … and 42.5 cm respectively were measured in these areas.
L 245, 274-276, 316-320, 327, 330, 375, 387, 389, 403, 412, 423-424. There is no space before the percentages.
L 243, 252. Please describe nearby representative areas. How they were determined?… the plots are identified in such a way as to be the most representative of the stands to which they belong.
L 246. What was the distance to SNFI plots? … the plots had to be located near plots in the 2005 SNFI.
L 259. Describe Assman dominant height.
L 260-263. The listed variables were not used in the analysis.
L 263. How was estimated the number of seedlings? What was the plot size for the seedling assessment?
L 304. The variables should be described in the title of the table.
L 322. Two-layered stands. It was not described that the stands were also classified by layer stratification.
L 338 (Figure 3) and L 381 (Figure 4). Correct the second SP mark on the figures to be the same as in the figure description.
L 346-352. Write in the plural.
L 387-388. Do not discuss the findings of other authors in the results section.
L 394 (Table 5) and L 397 (Table 6). Combine into one table.
Reviewer 2 Report
Dear Authors,
Please find the comments in the file attached
Regards
